# Certified robustness against physically-realizable patch attacks via randomized cropping

Wan-Yi Lin [1]   Fatemeh Sheikholeslami [1]   Jinghao Shi [2]   Leslie Rice [2]   Zico Kolter [1][2]

## Abstract

This paper proposes a certifiable defense against adversarial patch attacks on image classification. Our approach classifies random crops from the original image independently and classifies the original image as the majority vote over predicted classes of the crops. Leveraging the fact that a patch attack can only influence a certain number of pixels in the image, we derive certified robustness bounds for the classifier. Our method is particularly effective when realistic transformations are applied to the adversarial patch, such as affine transformations. Such transformations occur naturally when an adversarial patch is physically introduced in a scene. Our method improves upon the current state of the art in defending against patch attacks on CIFAR10 and ImageNet, both in terms of certified accuracy and inference time.

## 1. Introduction

Despite their incredible success in many computer vision tasks, deep neural networks are known to be sensitive to adversarial attacks; small perturbations to an input image can lead to large changes in the output. A wide range of defenses against adversarial attacks have been conducted in image classification, where the goal of the attacker is simply to change the predicted label(s) of an image (Kurakin et al., 2016; Szegedy et al., 2013; Madry et al., 2017). These works have mainly considered the so-called $\ell_p$-norm threat model, where an attacker is allowed to perturb the intensity at all pixels of the input image by a small amount. In contrast, *adversarial patch* attacks are considered as physically-realizable alternatives, modeling scenarios where a small object is placed in the scene so as to alter or suppress clas-

sification results (Brown et al., 2017). Here, the attack is spatially compact, but can change the pixel value to any value within the allowable range.

This paper develops a practical and provably robust defense against patch attacks. Inspired by the randomized smoothing defense (Cohen et al., 2019; Levine & Feizi, 2019) for the $\ell_p$-norm threat model, our approach classifies randomly sampled sub-regions or crops of an image *independently* and outputs the majority vote across these crops as predicted class of the input image. This approach is highly practical, as the crop classifier can be trained using standard architectures such as VGG (Simonyan & Zisserman, 2014) or ResNet (He et al., 2016) with random cropping as its data augmentation strategy. This is different from most existing work on certifiable defenses against patch attacks (Levine & Feizi, 2020; Xiang et al., 2020; Chiang et al., 2020) which need extra computation for certification during training. Also, the proposed approach separates the training procedure from the patch threat model, thus making the method flexible against realistic settings of patch attacks – the same crop classifier can be used to certify patches of the same size under different transformations such as rotation and aspect ratio changes, without having to train a different model for different transformations of the patch.

We summarize our main results on CIFAR10 and ImageNet in Table 1 in comparison with the current state of the art certifiable defense against patch attack (Xiang et al., 2020) with patch transformation. We report certified accuracy, which is the percentage of test images for which classification outcome equals to the ground truth label and is guaranteed to not change under patch attack. Our method is better in both speed and certified accuracy compared to De-randomized smoothing (Levine & Feizi, 2020) and PatchGuard (Xiang et al., 2020) under patch attack with possible affine transformations of the patch. In addition, our method outperforms these past approaches on ImageNet (though not on CIFAR10) in the setting where the patch aligns with coordinates of the image axes and does not undergo affine transformations as in Table 2, which was the setting considered in this past work.

We have made several contributions in this paper: first, we propose a defense against patch attack for image classifica-

---

[1]Bosch Center for Artificial Intelligence, Pittsburgh, PA, USA [2]Carneige Mellon University, Pittsburgh, PA, USA. Correspondence to: Wan-Yi Lin <wan-yi.lin@us.bosch.com>.

*Accepted by the ICML 2021 workshop on A Blessing in Disguise: The Prospects and Perils of Adversarial Machine Learning.* Copyright 2021 by the author(s).

| | CIFAR10 2.4% patch | | ImageNet 2.0% patch | |
|---|---|---|---|---|
| | certification acc. (clean acc.) | time in ms | certification acc. (clean acc.) | time in ms |
| Proposed method | **47.5** (**88.4**) | **0.7** | **12.2** (**55.7** ) | **21.8** |
| De-rand. smoothing | 17.5 (83.9) | 17.5 | 3.2 (43.1) | 703.2 |
| PatchGuard: De-rand. smoothing | 18.2 (84.5) | 18.2 | 3.5 (43.6) | 734.5 |
| PatchGuard: Bagnets | 27.1 (82.6) | **0.7** | 9.6 (54.4) | 25.7 |

*Table 1.* Worst-case certified accuracy (%), clean accuracy(%), and certification time of the proposed method, De-randomized smoothing (Levine & Feizi, 2020), and PatchGuard (Xiang et al., 2020) with De-randomized smoothing and Bagnets as base structure. For each method, we list the *worst* certified accuracy under affine transformation of the patch at test time. *Note that this is different from results in the original paper where patch transformations at test time are not considered.*

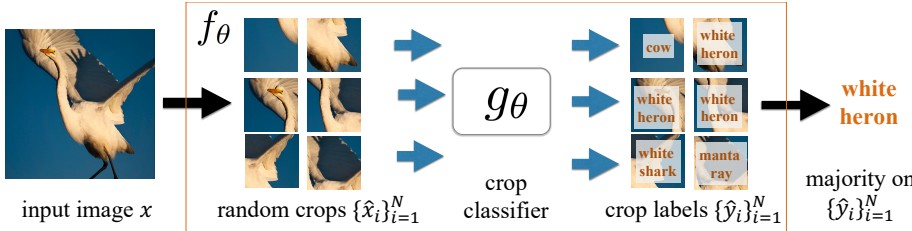

*Figure 1.* Forward pass of randomized crop defense

tion with certified robustness; second, the proposed method is fast in computing image certification and robust against patch transformation; third, the proposed method can be applied to any image classification model with only minimal changes to the training process.

## 2. Background and related work

Test-time adversarial attacks on ML models in general were studied in (Dalvi et al., 2004; Biggio et al., 2013), though the area gained considerable momentum when these methods were applied to deep learning systems to demonstrate that deep classifiers could be easily fooled by imperceptible changes to images (Szegedy et al., 2013; Goodfellow et al., 2014). This imperceptible attack is so-called $\ell_\infty$-norm attack, where attacks are permitted to modify any pixel in the image by (at most) some fixed amount. In this paper, we consider patch attacks (Brown et al., 2017; Eykholt et al., 2018), where a particular 'pattern' is designed to fool a deep learning system. There are also two threads of research in defending against patch attacks: 1) empirical defense strategies which show stronger empirical robustness but no analytical guarantees (McCoyd et al., 2020; Naseer et al., 2019; Hayes, 2018), and 2) certified defenses which provide analytical lower bounds of classification accuracy under patch attack. In Chiang et al. (2020), interval bound propagation was used to provide certification, but the method is not applicable to commonly used image classification networks such as ResNet50 or VGG16. De-randomized

smoothing (Levine & Feizi, 2020) uses ablation to exhaustively block all possible patch locations while (Zhang et al., 2020) uses Bagnets (Brendel & Bethge, 2019) with limited receptive field to contain the number of affected features. PatchGuard (Xiang et al., 2020) adds additional detection of patch location and then sets features extracted from the detected locations to be zero. We will compare our proposed method with De-randomized smoothing (DRS) and PatchGuard (PG) in the experiments and an overview of the two prior arts can be found in Appendix D.

## 3. Provable patch defense with randomized cropping

A physically-realizable adversarial patch attack can be found by solving the optimization problem:

$$\max_{\delta \in \Delta} \mathbf{E}_{(x,y) \sim D, t \sim T} \left[ \ell(f_\theta(A(x, \delta, t)), y) \right] \quad (1)$$

where $f_\theta : \mathcal{X} \to \mathcal{Y}$ denotes some hypothesis function; $\theta$ denotes parameters of the model, $x \in \mathcal{X}$ denotes the input to the network and $\delta \in \mathcal{X}$ the perturbation to the input; $\Delta \subseteq \mathcal{X}$ is the set of allowable perturbations; $y \in \mathcal{Y}$ denotes the true label; $\ell : \mathcal{Y} \times \mathcal{Y} \to R$ denotes a loss function that measures the performance of the image classifier; and $\mathcal{T} : \mathcal{X} \to \mathcal{X}$ is the set of transformations that the perturbation $\delta$ might go through. The feasible set $\Delta$ denotes a simple allowable set of values for the patch, which typically would just be constrained to lie in valid RGB space. Patches *overwrite* a

| | CIFAR10 | | | ImageNet | | |
|---|---|---|---|---|---|---|
| Patch Size | ours | DRS | PG+DRS / Bagnets | ours | DRS smoothing | PG+DRS / Bagnets |
| 0.4% | 65.7 | 68.9 | **69.2** / 53.2 | 24.7 | 22.3 | **24.8** / 23.1 |
| 1.0% | 60.2 | 62.7 | **65.3** / 41.2 | **20.1** | 17.7 | 19.9 / 18.6 |
| 2.0% | 55.2 | 60.9 | **61.1** / 37.2 | **16.4** | 14.0 | 16.0 / 13.3 |
| 2.4% | 52.3 | 57.1 | **58.1** / 31.7 | **15.3** | 13.1 | 14.7 / 11.2 |
| 3.0% | 37.8 | 42.1 | **43.5** / 25.1 | **14.2** | 11.2 | 13.01 / 8.9 |

*Table 2.* Certified accuracy (%) over CIFAR10 and ImageNet of our method, De-randomized smoothing (DRS) (Levine & Feizi, 2020), and PatchGuard (PG)(Xiang et al., 2020). We consider images with $p_c \geq 0.95$ be certified.

portion of the image with the patch perturbation itself, at a given location with a given set of transformations, such as scaling, rotation, and other transformations in $T$. We refer to this combination as the patch application function $A : \mathcal{X} \times \mathcal{X} \times \mathcal{T} \to \mathcal{X}$, where $A(x, \delta, t)$ denotes the application of patch $\delta$ to image $x$ with transformation $t$. Throughout this paper, we consider the patch being one connected area bounded by a rectangle with size $p_i \times p_j$.

### 3.1. Randomized cropping defense

Although a patch attack can change the pixel value to any arbitrary value, it can only influence the pixels within the patch itself. Therefore, networks extracting features with compact receptive fields and aggregating such local features for final classification are more robust against patch attacks. Based on above observations, we propose a *randomized cropping* approach as shown in Fig. 1: given a full-size test image $x$ as input, we first randomly select $n$ crops uniformly over all possible locations, where crop size $k_i \times k_j$ is smaller than image size $m_i \times m_j$ in both dimensions. Each of the sampled crops ($\widehat{x}_i$) then goes through the crop-based classifier $g_\theta$ and its predicted class ($\widehat{y}_i$) is obtained. The final classification of $x$ is the majority of $\{\widehat{y}_i\}_{i=1}^n$. The equivalent pseudo code of Fig. 1 is in Appendix E. We argue that the training procedure of randomized cropping classifier $f_\theta$ only requires training the crop classifier $g_\theta$ (see Section A) without any assumption on attack parameters (size, shape, and location), hence our method is robust against different patch shapes given the same patch size. We show supportive experimental results in Section 4 for this claim.

### 3.2. Certifiable robustness bound

Here we compute the probability that an image $x$ is certified robust, i.e., the classification outcome of $x$ cannot be changed by patch attack of given size. Given a clean image, we randomly sample $n$ crops $\{\widehat{x}_i\}_{i=1}^n$ and obtain the set of $n$ predicted classes $\{\widehat{y}_i\}_{i=1}^n$ of the crops. Let $n_1$ be the number crops that are predicted as the majority in $\{\widehat{y}_i\}_{i=1}^n$,

and $n_2$ be the number crops that are predicted as the second majority. If there are fewer than $n_{2to1} = \frac{\lfloor n_1 - n_2 \rfloor}{2} + 1$ crops that overlap with the patch, then $x$ is certified robust. Therefore, $p_c$ can be represented as the probability that fewer than $n_{2to1}$ crops out of the $n$ sampled crops overlap with the adversarial patch. The below derivation assumes that the random selection is uniform over all crop locations with replacement. Derivations for uniform sampling crops without replacement can be found in Appendix F.

Following the argument above, we first compute the probability that a single sampled crop $x_i$ overlaps the adversarial patch. This probability $p_a = \frac{n_{adv}}{n_{all}}$ can be computed as the total number of crops overlapping with the patch (denoted by $n_{adv}$) divided by the number of all possible crops ($n_{all}$). When the patch is at the center of the image where it can influence the highest number of crops,

$$n_{adv} = \min(p_i + k_i - 1, m_i - k_i + 1)$$
$$\times \min(p_j + k_j - 1, m_j - k_j + 1), \quad (2)$$
$$n_{all} = (m_i - k_i + 1) \times (m_j - k_j + 1).$$

The probability of certification $p_c$ is equal to the probability that out of the $n$ crops $\{\widehat{x}_i\}_{i=1}^n$, at most $n_{2to1} - 1$ of them overlaps with the adversarial patch:

$$p_c = \sum_{i=0}^{n_{2to1}-1} C_i^n * p_a^i * (1 - p_a)^{(n-i)}, \quad (3)$$

where $C_i^n$ is the binomial coefficient ($n$ choose $i$). If $p_c$ is close to 1.0, the input image is certifiably robust under patch attack. In the experiments we select $p_c \geq 0.95$. Throughout this paper, we consider *certified accuracy*, meaning only data with predicted class $y'$ being the ground truth class will be considered for certification. Note that because the crops are randomly sampled, $n_{2to1}$ and hence $p_c$ is an instance of a random distribution. We argue that when $n$ is large enough, there $n_{2to1}$ and $p_c$ will have very small variance. Experiments to verify this point are in Appendix G.

| | CIFAR10 2.4% patch | | | ImageNet 2% patch | | |
|---|---|---|---|---|---|---|
| Transformation | ours | DRS | PG+DRS / Bagnets | ours | DRS smoothing | PG+DRS / Bagnets |
| AR 1:1 | 52.3 | 57.1 | 58.1 / 31.7 | 16.4 | 14.0 | 16.0 / 13.3 |
| AR 2.7:1 | 50.7 | 65.8 | 67.2 / 30.4 | 15.8 | 15.2 | 17.6 / 12.4 |
| AR 6:1 | 47.5 | 71.1 | 74.5 / 27.1 | 12.2 | 17.9 | 19.0 / 9.6 |
| AR 1:2.7 | 50.7 | 40.6 | 42.4 / 30.4 | 15.8 | 11.3 | 11.9 / 12.4 |
| AR 1:6 | 47.5 | 17.5 | 18.2 / 27.1 | 12.2 | 3.2 | 3.5 / 9.6 |
| Rotate 45° | 48.0 | 50.3 | 52.1 / 28.1 | 12.4 | 12.1 | 12.8 / 10.1 |
| Worst case | **47.5** | 17.5 | 18.2/27.1 | **12.2** | 3.2 | 3.5/9.6 |

*Table 3.* Certified accuracy (%) under patch rotation and different aspect ratio at test time. We consider images with $p_c \geq 0.95$ be certified.

## 4. Experimental results

We conduct experiments on two benchmark datasets: CI-FAR10 and ImageNet. For both datasets we report certified accuracy as percentage of images that are classified correctly and can be certified with probability higher than 0.95. Detailed experiment setup are in Appendix B, and effects of different parameters are discussed in Appendix C.

**Clean accuracy and inference time.** Clean model accuracy and inference time per image are shown in Table 1. Our method is faster than both prior methods under all cases. This is because although multiple forward passes are needed, the number of crops is smaller than equivalent sub-regions in Bagnets and crops are much smaller the full images as in De-randomized smoothing (DRS). PatchGuard with Bagnets (PG-Bagnets) is slower than our method in ImageNet for having to go through logits of all classes sequentially.

### 4.1. Without patch transformation at test time

We first present the certification accuracy in Table 2 of our method along with De-randomized smoothing (DRS) (Levine & Feizi, 2020) and PatchGuard (PG)(Xiang et al., 2020) with De-randomized smoothing (PG+DRS) and Bagnets (PG+Bagnets) as base structure on patch size ranging from 0.4% to 3% when there's no patch transformation at test time, i.e., square patches in both test and train time. Methods with the highest certified accuracy are highlighted in bold. We can see that although our method is sub-optimal for CIFAR10, for ImageNet we have the highest certified accuracy except for patch size 0.4%. From Table 1 and 2 we can see that on ImageNet our proposed method is the fastest and with the highest certified accuracy as well as clean accuracy the certified accuracy – this shows that the proposed randomized cropping defense is practical in the aspects of fast certification and high certified accuracy for the dataset that is closer to real-life pictures.

### 4.2. With patch transformation at test time

As described in Section 3, one cannot assume such area would always align with coordinate axes of the image – even if the physical patch itself is square, when the scene is captured with different camera angles, the patch on the image will be rotated (rotation in x-y plane) or stretched (rotation in depth). Therefore it is important that the patch defense can also provide guarantees when the patch is rotated or if the aspect ratio is varied.

In Table 3 we compare our method with DRS and PG-DRS/PG-Bagnets when a square patch is rotated 45 degrees and with aspect ratio (AR) 1:1, 6:1, 2.7:1, 1:2.7, 1:6, respectively. We highlight the highest certified accuracy under worst transformation in the last row. Aspect ratio 1:1 without rotation is the same as Table 2 but listed here as reference. We can clearly see that in Table 3, with the same patch size, transformation brings down the certified accuracy of all three competing methods, however, the proposed randomized cropping defense has the highest certified accuracy under worst-case patch transformation. This is because of our *random sampling* strategy that we have neither fixed the locations of sub-regions as in Bagnets, nor fixed smoothing strategy as in De-randomized smoothing.

## 5. Conclusion

This paper proposes a new defense against adversarial patch attacks. The proposed approach decomposes an image into a random assortment of crops, each of which is processed by a classifier, and the majority across the crops is used as the classification outcome for the input image. This approach provides a significant advance in improving certified accuracy when patches could be transformed at test time, while maintaining a high clean accuracy compared to prior art.

## Acknowledgements

This work is funded by DARPA HR11002020006

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

# A. Training randomized cropping classifier

The robust classifier $f_\theta$ has three components: (1) sampling $n$ crops with uniform distribution, (2) classifying crops with $g_\theta$, and (3) majority voting. Since both random sampling of the crops via a uniform distribution and majority voting have no trainable parameters, the set of trainable parameters of $f_\theta$ is the same as $g_\theta$. Therefore training the robust classifier $f_\theta$ only involves training the crop classifier $g_\theta$, which can be trained in standard ways with only data augmentation of randomly cropping the input $x$. Other variables are either pre-determined hyperparameters (crop size $k_i$, $k_j$), or can be adjusted at test time (number of crops $n$).

# B. Experiment setup

For CIFAR10, we use ResNet9 without the final pooling layer as the crop model, and for ImageNet we use ResNet34 as the crop model and resize and pad the image to 224x224. Experiments and timing are done on single Nvidia 2080 Ti GPU, and 16-core Intel i7-5960X CPU. For both models we use cyclic learning rate with $10^{-3}$ initial learning rate. The models are trained 30 epochs and training time is 14 GPU-hours and 328 GPU-hours for CIFAR10 and ImageNet, respectively.

we chose crop size of 10x10 (10% total area) for CIFAR and 80x80 (13% total area) for ImageNet; for these experiments we assume patch shape is *square with no rotation*. For each input image in CIFAR10, we randomly sample 128 crops and for each image in ImageNet we sample 256 crops.

**Positional encoding.** Because both CIFAR10 and ImageNet images have a certain region of interest which is usually at the center of the image, some parts of the images contain more information than others. Therefore, crops sampled at different position of the image contain different information. To represent such information, we add learnable positional encoding (Vaswani et al., 2017) to the first layer of our classifier.

# C. Discussions on different parameter setting

We also show the certification accuracy with different thresholds for $p_c$ Table 4 in Appendix.

**How to choose crop size?** Assuming that the number of crops $n$ is fixed, then in general larger crops leads to a better clean performance, as each crop contains more information when it covers more pixel area. Also with larger crops, the crop classification accuracy of $g_\theta$ would be better, indicating that $n_{2to1}$ could be larger and increases the probability of certification $p_c$. However, larger crops also means that the probability of overlapping the adversarial patch is higher (Eq. 2) which will decrease $p_c$. Therefore, for a given size

of adversarial patch, there exists an optimal crop size which maximizes the certification probability.

To demonstrate the influence of crop size on certification accuracy and clean accuracy, clean and certified accuracy with or without positional encoding, with regards to different crop sizes are shown in Figure 2 for 2.4% patch on CIFAR10 and 2.0% patch on ImageNet – we use square crops and square patches aligning with coordinate axes of the image, i.e., no patch transformation. Comparing clean accuracy with and without positional encoding, we can see that although clean accuracy still increases as crop size increases when crop classifier includes positional encoding, but not as much as without positional encoding. Such results show positional encoding does provide extra information for crops sampled from different locations. On the other hand, comparing clean accuracy with certified accuracy, it is clear that the certified accuracy actually gets lower when crop size crosses some threshold as discussed above.

Note that similar experiments as in Figure 2 can be conducted for different sizes of adversarial patches to find optimal crop size. However we used fixed crop size in Table 1, 2, 3 to compare with DRS and PG to have a fair comparison, as these two approaches do not have tune-able ablation/kernel size.

**How to choose number of crops?** We show certified accuracy and inference time with different number of sampled crops for 2.4% patch on CIFAR10 and 2.0% patch on ImageNet. Certified accuracy of using all crops, i.e., selecting crops at all locations once without sampling, is plotted with the black dotted line as reference. This line can be viewed as the upper bound of the proposed method. With more crops sampled, the certified accuracy increases but however inference time also increases close to linearly. Therefore we chose relatively small number of crop samples to balance between inference time and certification accuracy.

# D. Overview of De-randomized smoothing (DRS) and PatchGuard (PG)

De-randomized smoothing (Levine & Feizi, 2020) ablates all possible parts of the image and aggregates logits of the ablated images for certification. The method compares number of ablated images with high logit value of the majority predicted class with the number of ablated images with high logit value of the second majority predicted class. If the difference between the two is larger than two times possible number of "ablation blocks" affected by adversarial patch, then this image is certified robust against patch attack. The paper proposes two modes of ablation: block smoothing and band smoothing. Block smoothing ablates square blocks while band smoothing ablates one column of the image. To represent ablated regions, the image classifier accept

|  | $p_c \geq 0.93$ | $p_c \geq 0.95$ | $p_c \geq 0.97$ | $p_c \geq 0.99$ |
|---|---|---|---|---|
| CIFAR10 2.4% | 52.8 | 52.3 | 52.0 | 49.6 |
| ImageNet 2.0% | 17.2 | 16.4 | 15.3 | 14.1 |

*Table 4.* Certified accuracy (%) with different thresholds for $p_c$ without patch transformation

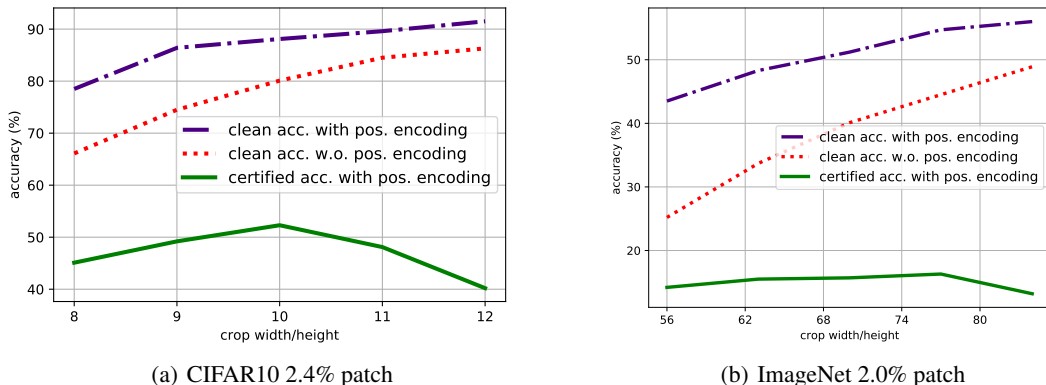

(a) CIFAR10 2.4% patch      (b) ImageNet 2.0% patch

*Figure 2.* Clean and certified accuracy with different crop sizes. Left: CIFAR10 with 2.4% patch. Right: ImageNet with 2.0% patch

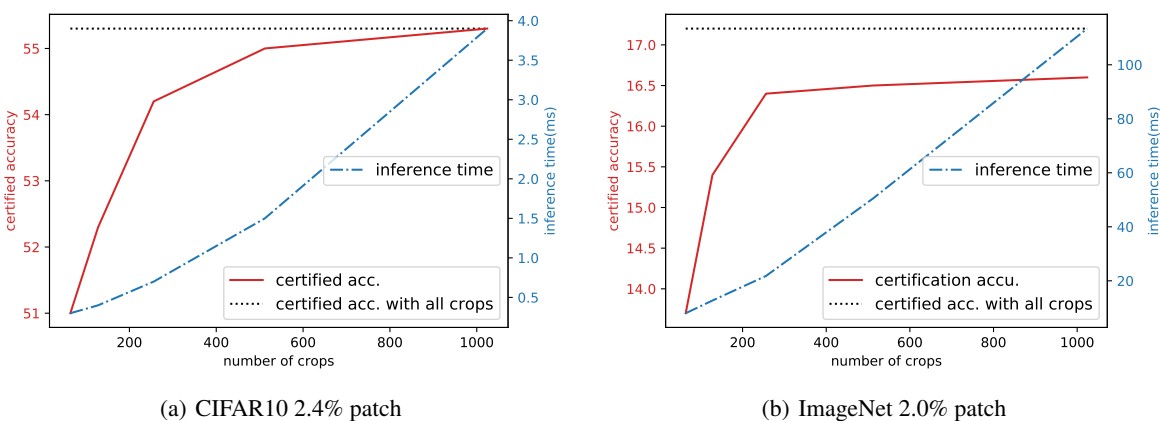

(a) CIFAR10 2.4% patch      (b) ImageNet 2.0% patch

*Figure 3.* Certified accuracy and inference time(ms) with different number of sampled crops. Left: CIFAR10 with 2.4% patch. Right: ImageNet with 2.0% patch

**Algorithm 1** Certify if patch attack can change the predicted class of an image $x$

---

**Input:** Full size image $x$, label $y$, crop classifier $g_\theta$
$i = 1$ **to** $n$ set $\widehat{x_i}$ as a $k_i \times k_j$ crop of $x$ at random location
$\widehat{y_i} = g_\theta(\widehat{x_i})$ //*Predicted class of crop* $\widehat{x_i}$
$\widehat{y'}, \widehat{y''}$ = majority and second majority of $\{\widehat{y_i}\}_{i=1}^n$,
$n_1, n_2$ = number of crops classified as $\widehat{y'}, \widehat{y''}$, respectively
**If** $\widehat{y'} \neq y$
**return** not certified
**Else**
compute $p_c$ using Eq. 3,
**if** $p_c$ is close to 1.0 **then** return certified,
**else** return not certified

---

three additional channels representing ablated pixels of the original RGB channels.

The main idea of PatchGuard (Xiang et al., 2020) is to detect possible patch location and mask these locations for downstream robust classification. To detect patch location, the method identify if there are any local regions that contribute abnormally strongly to a class. If there does exist such a region, it is considered as the potential location of a patch and the features extracted from the region is masked/discarded. Because such robust masking procedure can be combined with other robust classification approaches, the paper evaluates its robust masking with two classification models: De-randomized smoothing (PG-DRS) and BagsNet (PG-BN).

## E. Detailed certification procedure

We summarize the certification procedure for a single image in Algorithm 1. A more robust classifier should be able to certify and correctly classify a higher percentage of clean images in the test set.

## F. Uniform sampling without replacement

In this section we derive certification probability and experimental results with uniform sampling without replacement. Let $n_{all}^i$ be the number of all possible location for the $i^{th}$ sampled crop, $n_{adv}$ be the number of crop locations that overlaps with the adversarial patch, and $p_a^i$ is the probability that the $i^{th}$ crop overlaps with the adversarial patch, then

$$\begin{aligned}
n_{adv} &= \min(p_i + k_i - 1, m_i - k_i + 1) \times \\
&\quad \min(p_j + k_j - 1, m_j - k_j + 1), \\
n_{all}^i &= (m_i - k_i + 1) \times (m_j - k_j + 1) - i, \text{ and} \quad (4) \\
p_a^i &= min(1, \frac{n_{adv}}{n_{all}^i}).
\end{aligned}$$

The probability of the image being certified $p_c$ is then the probability of less than $n_{2to1}$ crops overlaps with the patch.

The closed-form expression of $p_c$ is complicated yet not informative and hence omitted here. Comparing $p_a^i$ with the $p_a$ in Section 3, we can see that when sampling without replacement, the probability of sampling a crop that overlaps with the adversarial patch increases with the number of crops sampled, and hence decreases the probability of certification $p_c$. On the other hand, sampling without replacement enlarges the expected area that crops would cover, so the clean performance will be better than sampling with replacement.

We compare the certified and clean accuracy with and without replacement in Table 5. As number of crops increases, we can see that the gain of clean accuracy for sampling without replacement decreases because when number of crops increases, even sampling with replacement is likely to cover most of the pixels, and the gain is slightly more significant in ImageNet than CIFAR10. This may be because ImageNet images are in general more complex than CIFAR10 and having the crops covering more pixels over the image could help the overall classification. The certified accuracy for sampling without replacement gets worse than with replacement since the probability of sampling a crop increases more significantly as the number of crops sampled increases. This is particularly true for CIFAR10 – with image size 32x32 and patch size 10x10, the number of non-overlapping crops that does not overlap with the patch is only 484, out of 1024 all possible locations. This means when sampling 512 crops, there are at least 28 crops overlapping with the adversarial patch, which significantly decrease $p_c$.

## G. Experiments on $p_c$ interval

In this section we show the interval of $n_{2to1}$ and $p_c$ with different number of crops sampled. We run the certification process over test set 200 times to obtain the interval and variance of $n_{2to1}$ and $p_c$ for each test image. Interval is defined as difference between the highest value and the lowest value. We ran this experiment over patch size 2.4% for CIFAR10 and 2.0% for ImageNet.

As shown in Table 6, with increasing number of crops, both interval and variance of $n_{2to1}$ and $p_c$ decrease, to negligible values.

| num. of crops | CIFAR10 2.4% | | ImageNet 2.0% | |
|---|---|---|---|---|
| | clean (R/NR) | certified (R/NR) | clean (R/NR) | certified (R/NR) |
| 64 | 88.3/88.6 | 51.0/51.1 | 50.1/50.4 | 13.7/13.9 |
| 128 | 89.6/89.7 | 52.3/45.3 | 54.7/54.9 | 15.4/15.5 |
| 256 | 89.8/89.8 | 54.2/24.7 | 54.8/54.8 | 16.4/16.2 |
| 512 | 89.8/89.8 | 55.0/0.8 | 55.0/55.0 | 16.5/16.2 |

*Table 5.* Certified and clean accuracy (%) with (R) / without replacement (NR) over CIFAR10 and ImageNet vs number of crops without patch transformation. Patch size is 2.4% for CIFAR10 and 2.0% for ImageNet and crop size is 10x10 for CIFAR10 and 80x80 for ImageNet. We consider images with $p_c \geq 0.95$ to be certified.

| num. of crops | CIFAR10 2.4% | | ImageNet 2.0% | |
|---|---|---|---|---|
| | $n_{2to1}$ interval/variance | $p_c$ interval/variance | $n_{2to1}$ interval/variance | $p_c$ interval/variance |
| 64 | 1.3/0.6 | 1.9/0.9 | 2.1/0.8 | 3.0/1.0 |
| 128 | 0.8/0.4 | 0.8/0.5 | 1.4/0.5 | 1.5/0.7 |
| 256 | 0.5/0.2 | 0.5/0.3 | 0.9/0.3 | 0.8/0.4 |
| 512 | 0.2/0.1 | 0.2/0.1 | 0.6/0.1 | 0.5/0.1 |

*Table 6.* Averaged interval and variance of $n_{2to1}$ and $p_c$ (in $10^{-2}$) vs number of crops without patch transformation. Patch size is 2.4% for CIFAR10 and 2.0% for ImageNet and crop size is 10x10 for CIFAR10 and 80x80 for ImageNet.