# OpenReview forum: "Certified robustness against adversarial patch attacks via randomized cropping"
_ICML.cc/2021/Workshop/AML — ICML 2021 Workshop AML Oral_

### Official Review · Reviewer_J8hY · 2021-06-19
**A novel idea for improving certified robustness against patch attacks**

**Rating:** Accept
**Confidence:** 4

**Review:**

This paper proposed an interesting idea that patch attacks can be certifiably defended by random cropping. The idea of randomized cropping comes from that randomized smoothing is an effective way for l-2 attacks. The author generalized the idea to patch attacks that only modify certain pixels. Both the theoretical and experimental results are effective.

---

### Decision · Program_Chairs · 2021-06-21

**Decision:**

Accept (Oral)

**Comment:**

A good paper for certifying robustness against patch attacks. The idea of randomized cropping is a natural extension of randomized smoothing.